# Differences in Aerobic Fitness between an Obese Adolescent with Prader-Willi Syndrome and Other Obese Adolescents and Exercise Training Results

**DOI:** 10.3390/ijerph17051496

**Published:** 2020-02-26

**Authors:** Yentung Su, Hungya Huang, Shanghui Tuan, Minhui Li, Kolong Lin

**Affiliations:** 1Department of Physical Medicine and Rehabilitation, Kaohsiung Veterans General Hospital, Zuoying District, Kaohsiung 813, Taiwan; ph860122@gmail.com (Y.S.); hhya@vghks.gov.tw (H.H.); mhli@vghks.gov.tw (M.L.); 2Department of Rehabilitation Medicine, Cishan Hospital Ministry of Health and Welfare, Cishan District, Kaohsiung 84247, Taiwan; shtuan@vghks.gov.tw

**Keywords:** Prader-Willi syndrome, cardiopulmonary exercise test, adolescent

## Abstract

Prader-Willi syndrome (PWS) is a genetic disorder characterized by specific physical and behavioral abnormalities and considered the most commonly known genetic cause of morbid obesity in children. Recent studies indicate that patients suffering from this syndrome have significant problems in skill acquisition, muscle force, cardiovascular fitness, and activity level. In this study, we report an obese adolescent PWS patient of poor aerobic fitness compared with 13 obesity adolescents, and great improvement in cardiopulmonary exercise test (CPET) outcomes of the PWS patient measured after two weeks of physical exercise training programs.

## 1. Case Report

A 17 year old patient with PWS was referred for exercise training programs for poor weight control and obesity. The relevant medical history included PWS, precocious puberty, hypercholesterolemia, hyperuricemia and scoliosis.

His body weight had increased from 80 to 120 kg in two years due to poor diet control. He had intentional weight loss from 120 to 106 kg in one month before admission.

After admission, elevated blood pressure and dyslipidemia was noted. Abdominal sonogram showed fatty liver. Electrocardiograph (ECG) showed sinus rhythm. Heart echogram showed normal cardiac function. Laboratory investigations indicated normal electrolytes with raised GOT 41 U per liter and GPT 88 U per liter. His baseline body composition, before exercise training programs, is given in Table 1.

During hospitalization, daily medications included atenolol 50 mg and atorvastatin 20 mg for blood pressure and dyslipidemia control. Of note, pre-exercise training CPET was done before taking atenolol, meanwhile, post-exercise training CPET was done after taking two weeks of atenolol 50 mg/day.

He then received body composition measurement and CPET. Vector bioelectric impedance analysis (VBIA) was used to measure his body composition. We performed VBIA with the bioelectrical impedance vector analysis software by using the resistance–reactance graph method. Zeus 9.9 PLUS (Jawon Medical Co. Ltd., Kungsang Bukdo, South Korea) was used to analyze his body composition.

Before CPET, he was familiarized with the procedures and equipment used in the test via a demonstrative explanation. A symptom-limited exercise testing, which consisted of a treadmill, a flow module, a gas analyzer, and an electrocardiographic monitor (Metamax 3B, Cortex Biophysik GmbH Co., Leipzig, Germany), was used to measure his exercise capacity.

He underwent exercise testing according to the ramped Bruce protocol, as per suggestions from The American College of Sports Medicine (ACSM). The test was terminated when he demonstrated subjective unbearable symptoms, when he could no longer continue, or when he attained maximal effort as indicated by the ACSM.

VO_2_ and carbon dioxide production (VCO_2_) were measured during the test by using the breath-by-breath method. Minute ventilation (VE), blood pressure (BP), and heart rate (HR) were also measured. HR recovery (HRR) was calculated as the difference between HR at 1 min after testing and maximum HR during testing. Metabolic equivalent (MET), which is considered equivalent to 3.5 mL of oxygen per kilogram of body mass per minute, was calculated after measuring VO_2_. The anaerobic threshold was determined by VE/VO_2_ and VE/VCO_2_ methods. The following physiological criteria for reaching VO_2_max were used: (1) respiratory exchange ratio (RER) > 1.1, (2) peak HR > 200 bpm, and (3) HR > 85% of age-predicted maximum.

The results were given in Table 2 as before exercise training programs. He did not reach VO_2_max due to intolerant dyspnea.

We prescribed an exercise training program according to his CPET results.

His exercise prescription included daily aerobic exercise training and more than 3 d wk^−1^ of anaerobic exercise training as part of 60 min∙d^−1^ exercise.

For the aerobic exercise training, the training intensity was determined by the target VO_2_ AT and heart rate reserve method, as described in the study by Scharff-Olson et al. [1], at 60%–80% of the participant’s heart rate reserve. We chose 70% of his heart rate reserve, so the target heart rate was (149 − 93) × 0.7 + 93 = 132 beat/min. His VO_2_ AT was 2.8 MET. The target VO_2_ AT would be (2.8 − 1) × 0.7 + 1 = 2.26 MET = 7.91 mL/kg/min.

According to the target 2.26 MET, step exercise with up-stair and down-stair was recommended.

For anaerobic exercise training, he was given moderate fatigue exercise with thoracic expansion exercise (including side extension, trunk extension), and core muscle training (including superman, leg raise, plank and crunch). He performed 8–15 submaximal repetitions of the exercise.

After 2 weeks of exercise training programs, he did CPET again and all aspects of performance improved, with an increase in VO_2_ and workload at all stages of testing. These changes are given in Table 2. However, his body composition showed minor improvements in body weight and body fat, which are given in Table 1. We prescribed a home exercise training program according to the post-exercise training CPET results, and he was discharged from hospital.

To compare with other obese adolescents, we collected 13 obese patients from our hospital’s CPET database as a control group. Inclusion criteria were a BMI above 33, male gender and age between 15 to 18 years old. Among them, there are five people without any underlying disease, two people with Kawasaki disease and six people with congenital heart disease (including VSD, TGA or PSVT). The results are shown in Table 3. Even with these underlying diseases, the control group’s aerobic fitness (AT MET, peak MET) is still markedly better than that of the PWS patient.

This study was approved by the Institutional Review Board at the Kaohsiung Veterans General Hospital (identification number: VGHKS17-CT11-11).

## 2. Discussion

This 17 year old boy with PWS had increased his weight from 80 kg to 120 kg in two years due to poor diet control. He had intentional weight loss from 120 to 106 kg one month before exercise training programs. He was admitted for further weight control and exercise training.

During hospitalization, he took a small dose of atenolol 50 mg/day for two weeks to control blood pressure. The pre-exercise training CPET was measured before taking the medicine, so the comparison with other patients would not be affected. However, the post-exercise training CPET results might showed reduced heart rate accuracy, but maximal oxygen consumption or maximal workload is not affected [2,3].

After two weeks of exercise training programs, he had only 3 kg weight loss and fat composition showed no improvement (see Table 1). The results are corresponding to previous studies [4,5], in which body fat was only reduced by 2.5% to 3.6% after 20 weeks of concurrent and aerobic training. In another similar study [6], body fat was reduced by 1.1% to 1.6% after 6 months training. These results illustrate the difficulty for body fat to decrease after 2 weeks of exercise training programs.

There are possible explanations for the marked improvement in aerobic fitness on CPET results. First, his aerobic capacity could be truly improved. For non-obese adult PWS patients, six month walking programs could significantly improve their aerobic capacity [7]. Second, he could have improved his body control, such as exercise skill, breathing muscle strength and breathing pattern, and these factors contributed to the better aerobic exercise performance. There is another study of PWS patients that showed significant improvement in knee flexor strength and ankle dorsal flexion after 2 weeks of training [8].

When compared to other, similar obese adolescents, the PWS patient showed much declined aerobic fitness (See Table 3). BMI was significantly negatively associated with both AT MET and peak MET [9]. However, the PWS patient still showed much declined AT MET and peak MET even though his age, gender and BMI were similar to the control group. The results are similar to another study of adult PWS patients [10]. PWS patients might be also affected by additional factors besides obesity, such as blunted parasympathetic nervous system reactivation, delayed sympathetic withdrawal and poor cardiovascular fitness [11].

Due to the relatively declined aerobic fitness, the designs and prognosis of exercise training programs for PWS patients should be different from those of ordinary obese patients. Exercise prescription based on direct measurement of peak VO_2_ could be more suitable for the PWS patients’ condition. Muscle control and breathing techniques could be manifested. Specific rehabilitation training programs for PWS patients have previously shown an effect on BMI [8]. It is believed that with CPET evaluation, the specific exercise training programs for PWS could be more individualized, adaptable and efficient. However, further research, including more cases, is needed.

## 3. Conclusions

This case of an obese adolescent with Prader-Willi syndrome showed much declined aerobic fitness compared to the control group, even though age, gender and BMI were similar. The case also showed great improvement in aerobic fitness on CPET results after 2 weeks of exercise training programs.

The results demonstrate that the designs and prognosis of exercise training programs for PWS patients should be different from those of ordinary obese patients. Meanwhile, PWS patients could benefit from CPET for aerobic fitness evaluation, setting up exercise programs and following up.

## Figures and Tables

**Table 1 ijerph-17-01496-t001:** Body composition before and after exercise training program.

Variables	Before	After
Height (centimeters)	168	168
Body weight (kilograms)	106.1	104.4
BMI	37.6	36.9
Body fat (%)	43.1	43.6
FMI	16.2	16.1
FFMI	21.4	20.9
Rest SBP/DBP (mmHg)	128/89	112/90
Rest HR (bpm)	93	90
MVV (liters)	53.2	61.6

BMI, body mass index; FMI, fat mass index; FFMI, fat-free mass index; SBP, systolic blood pressure; DBP, diastolic blood pressure; HR, heart rate; MVV, maximal voluntary ventilation.

**Table 2 ijerph-17-01496-t002:** Cardiopulmonary exercise test (CPET) results before and after exercise training programs.

Variables	Before	After
SBP/DBP max (mmHg)	164/97	142/91
HR max (bpm)	149	133
MVV (liters)	53.2	61.6
HRR (beats)	23	22
VO_2_/WR slope (cc/min/watt)	4.9	5.9
PETCO_2_ rest (liter)	39	35
PETCO_2_ Max (liter)	46	44
**Anaerobic threshold**		
Time (minute: second)	3:45	4:45
Workload (MET)	2.8	3.6
VE (liter)	26.0	32.7
VO_2_ (ml/kg/min)	12.0	14.2
RER	0.82	0.81
HR (bpm)	132	116
**Peak exercise**		
Time (minute: second)	4:45	6:45
Workload (MET)	3.4	4.6
VE (liter)	37.1	60.0
VO_2_ (ml/kg/min)	14.3	17.8
RER	0.98	1.12
HR (bpm)	149	132

SBP, systolic blood pressure; DBP, diastolic blood pressure; HR, heart rate; MVV, maximal voluntary ventilation; HRR, heart rate recovery; VO_2_, oxygen consumption; WR, work rate; PETCO2, end-tidal carbon dioxide; MET, metabolic equivalent of task; VE, minute ventilation; RER, respiratory exchange ratio.

**Table 3 ijerph-17-01496-t003:** CPET results of the obese Prader-Willi syndrome (PWS) patient and other 13 obese patients in our hospitals.

Variables	PWS	Other 13 Obese Patients		
		Mean	SD	95% CI
BMI	37.6	36.25	2.62	34.66–37.83
rest SBP (mmHg)	* 128	143.00	18.43	131.86–154.14
rest DBP (mmHg)	* 89	79.92	9.84	73.97–85.87
rest HR (bpm)	* 93	78.69	11.80	71.56–85.82
AT MET	* 2.8	4.38	0.98	3.79–4.97
AT HR (bpm)	132	130.46	15.82	120.90–140.02
peak MET	* 3.4	6.35	1.14	5.66–7.04
peak HR (bpm)	* 149	165.54	16.63	155.49–175.59
peak VE (liters)	* 37.1	57.67	12.94	49.85–65.49
peak SBP (mmHg)	164	179.23	26.97	162.93–195.53
peak DBP (mmHg)	97	88.15	16.95	77.91–98.39
HRR (beats)	23	22.40	8.46	17.29–27.51

SD, Standard deviation; CI, confidence interval; BMI, body mass index; SBP, systolic blood pressure; DBP, diastolic blood pressure; HR, heart rate; AT, anaerobic threshold; MET, metabolic equivalent of task; HR, heart rate; VE, minute ventilation; HRR, heart rate recovery. * Asterisk represents that the data is out of the 95% confidence interval.

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
