# Peer review of "Differences in Aerobic Fitness between an Obese Adolescent with Prader-Willi Syndrome and Other Obese Adolescents and Exercise Training Results"

_ijerph, 2020, doi:10.3390/ijerph17051496_

Round 1

Reviewer 1 Report

Case report with low interest. Well written. Little reference is made to the comparison with groups of obese adolescent patients without Prader Willi.

Interest for journal readers: Medium-Low

Reviewer 2 Report

Su et al report the beneficial effect of an exercise training for a  patient with Prader-Willi syndrom.

The title of the manuscript is missleading, as no other patients were tested. The title should be adapted accordingly.

The manuscript is not easy to read and needs extensive editing of the English language/ style.

The authors state that the patient was treated with atenolol. Were all CPET performed under beta-blocker therapy? This should be  stated and also the timing after initiation of therapy.

Helpful would be a table that  includes baseline values in addition to CPET results. Units should be included in the tables.  For clearity all CPET date should be merged onto one table and another one for body composition data.

The discussion section would benefit from a more concise presentation.

line 45:  HR reserve (HRR) should be HR recovery as stated in table 2

table 1+2:

·         Do results presented under the heading "before/after exercise training program" and "before/after rehabilation" represent data from the same time periods?

·         CPX = CPET?

The reference Scharff-Olson et al is missing.

Reviewer 3 Report

The main issue here is that authors are comparing two different aspects, the PWS syndrome and ordinary obese children, why?

the manuscript is very interesting and very well writen, the question is that you are comparing two different things, obesity and PWS. they have different needs, the results found in pws can't be assumed as favourable to obesity.

Round 2

Reviewer 1 Report

You must review the references. References should be described as follows, depending on the type of work:

Author 1, A.B.; Author 2, C.D. Title of the article. Abbreviated Journal Name YearVolume, page range.

Reviewer 2 Report

the manuscript still needs extensive editing of English language/style

Reviewer 3 Report

The authors have explained their reasons and perspective and given the changes and corrections in the manuscript it could be published.
